# MIF-Modulated Spinal Proteins Associated with Persistent Bladder Pain: A Proteomics Study

**DOI:** 10.3390/ijms25084484

**Published:** 2024-04-19

**Authors:** Shaojing Ye, Nilesh M. Agalave, Fei Ma, Dlovan F. D. Mahmood, Asma Al-Grety, Payam E. Khoonsari, Lin Leng, Camilla I. Svensson, Richard Bucala, Kim Kultima, Pedro L. Vera

**Affiliations:** 1Research & Development, Lexington VA Health Care System, Lexington, KY 40502, USA; shaojing.ye@uky.edu (S.Y.); feima74@hotmail.com (F.M.); dlovan.kurdi@uky.edu (D.F.D.M.); 2Department of Medical Sciences, Clinical Chemistry, Uppsala University, SE-751 85 Uppsala, Sweden; nilesh.agalave@medsci.uu.se (N.M.A.); asma.al-grety@akademiska.se (A.A.-G.); payam.emami@medsci.uu.se (P.E.K.); kim.kultima@medsci.uu.se (K.K.); 3Department of Internal Medicine, Yale University, New Haven, CT 06510, USA; lin.leng@yale.edu (L.L.); richard.bucala@yale.edu (R.B.); 4Department of Physiology and Pharmacology, Karolinska Institutet (KI), SE-171 65 Solna, Sweden; camilla.svensson@ki.se; 5Department of Physiology, University of Kentucky, Lexington, KY 40506, USA

**Keywords:** persistent bladder pain, spinal proteins, macrophage migration inhibitory factor, CD74, CXCR4

## Abstract

Bladder pain is a prominent symptom in Interstitial Cystitis/Bladder Pain Syndrome (IC/BPS). We studied spinal mechanisms of bladder pain in mice using a model where repeated activation of intravesical Protease Activated Receptor-4 (PAR4) results in persistent bladder hyperalgesia (BHA) with little or no bladder inflammation. Persistent BHA is mediated by spinal macrophage migration inhibitory factor (MIF), and is associated with changes in lumbosacral proteomics. We investigated the contribution of individual spinal MIF receptors to persistent bladder pain as well as the spinal proteomics changes associated with relief of persistent BHA by spinal MIF antagonism. Female mice with persistent BHA received either intrathecal (i.t.) MIF monoclonal antibodies (mAb) or mouse IgG1 (isotype control antibody). MIF antagonism temporarily reversed persistent BHA (peak effect: 2 h), while control IgG1 had no effect. Moreover, i.t. antagonism of the MIF receptors CD74 and C-X-C chemokine receptor type 4 (CXCR4) partially reversed persistent BHA. For proteomics experiments, four separate groups of mice received either repeated intravesical scrambled peptide and sham i.t. injection (control, no pain group) or repeated intravesical PAR4 and: sham i.t.; isotype IgG1 i.t. (15 μg); or MIF mAb (15 μg). L6-S1 spinal segments were excised 2 h post-injection and examined for proteomics changes using LC-MS/MS. Unbiased proteomics analysis identified and relatively quantified 6739 proteins. We selected proteins that showed significant changes compared to control (no pain group) after intravesical PAR4 (sham or IgG i.t. treatment) and showed no significant change after i.t. MIF antagonism. Six proteins decreased during persistent BHA (V-set transmembrane domain-containing protein 2-like confirmed by immunohistochemistry), while two proteins increased. Spinal MIF antagonism reversed protein changes. Therefore, spinal MIF and MIF receptors mediate persistent BHA and changes in specific spinal proteins. These novel MIF-modulated spinal proteins represent possible new targets to disrupt spinal mechanisms that mediate persistent bladder pain.

## 1. Introduction

Protease activated receptors (PAR) are G-protein coupled receptors that are activated by endogenous peptidases to release their own ligands and thereby activate cell/intracellular signaling [1,2,3]. Four PAR receptors have been identified (PAR1 through PAR4 [3]), and the urothelium contains all four PAR receptors [4]. We previously reported that a single intravesical stimulation of PAR1 or PAR4 receptors using PAR1 or PAR4-activating peptide (PAR4-AP) in female mice resulted in a decrease in the lower abdominal mechanical threshold (an index of bladder hyperalgesia (BHA) or bladder pain) 24 h post-activation with little or no histological evidence of bladder inflammation, no change in bladder inflammatory markers, and no change in micturition parameters [5], a finding that has been replicated by others [6]. Furthermore, repeated (3×) activation of intravesical PAR4 receptors resulted in BHA that persisted for up to 9 days with minimal or no bladder inflammatory changes [7]. We used this approach as a model to study persistent bladder pain.

Interstitial Cystitis/Bladder Pain Syndrome (IC/BPS) is a chronic and debilitating condition characterized by pelvic pain (often referred to as the bladder) and other lower urinary tract symptoms [8,9,10]. Clinically, patients are often divided into two categories: those with Hunner lesions and a robust bladder inflammatory state, and those without Hunner lesions or any obvious bladder pathology [11,12]. This second group accounts for most (up to 90%) of IC/BPS patients [11]. These two groups of patients do not differ in their pain symptoms [13]. We used our rodent model of persistent bladder pain in mice to study the non-bladder-centric mechanisms responsible for persistent bladder pain as a way to understand and possibly treat bladder pain in non-Hunner lesion IC/BPS.

We hypothesized that spinal mechanisms were involved in maintaining persistent bladder pain in our model. In fact, we recently described that persistent BHA is associated with changes in lumbosacral proteomics [14]. Moreover, we determined that macrophage migration inhibitory factor (MIF) plays a crucial role in persistent BHA, as systemic treatment with a MIF antagonist has been found to completely reverse persistent BHA [7]. MIF is a pro-inflammatory cytokine that was first identified in 1966 [15] and is now widely recognized as a mediator of inflammation [16,17,18]. Recent evidence implicates MIF in mediating pain as well [19,20,21]. We were able to localize the effect of MIF to a spinal site, as lumbosacral MIF antagonism with a single intrathecal administration of MIF monoclonal antibodies (mAb) as an antagonist profoundly (albeit temporarily) reversed persistent BHA [22].

The aim of the current study was to extend our previous findings by identifying the spinal proteomics changes associated with MIF antagonism that alleviates bladder pain. In order to accomplish this, we first examined the optimal time after intrathecal lumbosacral anti-MIF treatment for bladder pain relief. Because MIF activates several receptors [23,24], a separate set of experiments examined whether intrathecal antagonism of specific MIF receptors (CD74; CXCR4) was effective in reversing persistent BHA. Then, using an unbiased proteomics approach, we determined the effect of lumbosacral MIF antagonism on spinal proteomics changes associated with persistent bladder pain.

## 2. Results

### 2.1. Experiment 1A: Time Course of Analgesic Effect on Persistent BHA after Intrathecal Treatment with Anti-MIF mAb

#### 2.1.1. Optimal Analgesic Effect on Persistent BHA after Intrathecal Treatment

Repeated (3×) intravesical instillation with PAR4-AP (PAR4-Activating peptide) resulted in a sustained decrease in the 50% von Frey threshold to lower abdominal stimulation, which we use as an indicator of bladder pain (Figure 1, y-axis). The 50% threshold temporarily recovered after the first instillation (Figure 1, Day 2), then decreased again after subsequent instillations and remained decreased for the remainder of the protocol (days 5–9; Figure 1A). We note that a lumbosacral intrathecal injection of isotype control had no effect on persistent pain during the observation period (6 h; Figure 1A; see inset). On the other hand, intrathecal injection with a monoclonal antibody to MIF (to neutralize spinal MIF) temporarily reversed persistent bladder pain (Figure 1B). A significant improvement in the 50% threshold (increased threshold) was observed at 1 h, while a maximal effect occurred at 2 h and then decreased at 4 and 6 h, respectively (Figure 1B; see inset). Thus, this experiment corroborates our previous findings that intrathecal MIF antagonism can ameliorate persistent bladder pain [22] and allows us to identify the peak analgesic effect at 2 h post-injection.

#### 2.1.2. Effects on Micturition Parameters

Awake micturition parameters (micturition frequency and volume) were measured on day 9, two days after intrathecal treatments. Mice receiving intrathecal mouse IgG1 had a micturition volume of 225 µL (±33.9), while mice receiving intrathecal MIF mAb had a mean micturition volume of 183 µL (±32.2). This difference was not statistically significant (*p* = 0.4; Figure 2A). Similarly, there were no significant differences in micturition frequency between the two groups (IgG1 = 2.2 (±0.75) micturition/3 h; MIF mAb = 2.2 (±0.75); *p* = 1; Figure 2B).

#### 2.1.3. Effects on Bladder Histology

We examined histological evidence of bladder inflammation and edema on day 9 of persistent BHA for the two groups in this study. Representative H&E-stained bladder sections from mice in the groups are presented in Figure 3. Histological examination revealed little or no bladder edema or inflammation in mice treated with intrathecal mouse IgG1 (Figure 3A) or intrathecal MIF mAb (Figure 3B).

Figure 4 shows a boxplot of the median and interquartile range along with a dotplot of the inflammation and edema scores for both groups. Differences between the two groups were evaluated with Fisher’s exact test. Figure 4A shows inflammation scores for mice treated with intrathecal mouse IgG1 and MIF mAb. Mice treated with intrathecal mouse IgG1 had a median inflammation score of 0.25 (IQR = 0), while the group treated with intrathecal MIF mAb had a median inflammation score of 0.38 (IQR = 0.25). This difference was not statistically significant (*p* = 0.52). In addition, edema scores (Figure 4B) between the two groups were not statistically different, with mice receiving intrathecal mouse IgG1 having an edema score of 0.25 (IQR = 0.19) compared to 0.50 (IQR = 1.1) for mice treated with intrathecal MIF mAb (*p* = 0.71).

### 2.2. Experiment 1B: Effects of Intraspinal Antagonism of Specific MIF Receptors on Persistent BHA

We examined the effects of intrathecal treatment with antagonists to specific MIF receptors in order to determine their role in mediating persistent BHA. We observed that intrathecal treatment with a monoclonal antibody to CD74 resulted in a small partial reversal of persistent BHA (increased threshold) on day 7, with a statistically significant peak effect at 4 h (Figure 5A; see CD74 inset). Intrathecal treatment on day 9 with a non-specific IgG2b had a small effect in that it further decreased (not increased) the 50% threshold (Figure 5A; see IgG inset). Intrathecal treatment on day 7 with AMD3100, a CXCR4 antagonist, resulted in partial reversal (increased threshold) of persistent BHA and a statistically significant peak effect at 4 h (Figure 5B; see AMD3100 inset), while intrathecal treatment on day 9 with PBS (vehicle control) had no effect on persistent BHA (Figure 5B; see PBS inset).

In addition, we compared the effect between all three intrathecal antagonists using and ANOVA and Dunnett’s post hoc tests. This analysis showed that treatment with CD74 mAb and/or AMD3100 produced significantly smaller effects than treatment with anti-MIF mAb (Figure 5C).

### 2.3. Experiment 2A: Novel MIF-Modulated Spinal Proteins in Persistent Bladder Hyperalgesia Model

In unbiased global proteomics analysis, we identified and relatively quantified 6739 lumbosacral proteins. We further narrowed our analyses to focus on changes in lumbosacral proteins *specific to MIF antagonism*, and thereby associated with relief from persistent BHA. We used comparisons between results from the experimental groups listed above (Methods; Section 4.3) and filtered protein changes so as to select proteins that showed the following:1.*Significant changes* after intravesical PAR4-AP and sham i.t. (unalleviated pain group; Group 2 above) compared to intravesical scrambled peptide treatment + sham i.t. (no pain group; Group 1 above);2.*Significant changes* after intravesical PAR4-AP + IgG1 i.t. (Group 3 above; pain with ineffective treatment) vs. Group 1;3.*No significant change* after intravesical PAR4-AP + MIF mAb i.t. (Group 4 above; treatment group) vs. Group 1.

This selection process controlled for nonspecific immune effects (from isotype IgG) and focused only on MIF-mediated spinal changes associated with MIF-mediated relief from persistent BHA. Figure 6 shows the selected proteins and the changes for each treatment. All comparisons are presented as ratios of protein levels (log2 scale) and all comparisons are normalized to the control (no pain group) levels.

Several lumbosacral proteins are associated with MIF-mediated relief from persistent BHA (Figure 6). Metallothionein-1 (MT-1), Metallothionein-2 (MT-2), V-set transmembrane domain-containing protein 2-like (VSTM2L), Xaa-Pro aminopeptidase 3, Mitochondrial 10-formyltetrahydro-folate dehydrogenase, and Methyl-CpG-binding domain protein 3 all showed decreases in spinal levels during persistent BHA compared to the no pain group, and this effect was reversed by intraspinal MIF antagonism (Figure 6A). Neuronal vesicle trafficking-associated protein 1 (NEEP21) and Peptidyl-tRNA hydrolase ICT1, on the other hand, showed an increase in spinal level after persistent BHA that was reversed by intraspinal MIF antagonism (Figure 6B). These findings support our hypothesis that spinal protein changes are associated with persistent BHA and that some of these changes are mediated by MIF at the level of the spinal cord.

### 2.4. Experiment 2B: Preliminary Validation of Specific Proteomics Targets

We replicated the experimental paradigm as described for experiment 1A: Group 1: Intravesical scrambled peptide + sham i.t. (no pain; n = 4); Group 2: Intravesical PAR4-AP + sham i.t. (persistent BHA; unalleviated pain group; n = 4). This setup was then used to examine VSTM2L levels in the L6/S1 cord using immunohistochemistry.

In the control no-pain group (Figure 7A), VSTM2L staining (Santa Cruz; sc376538; 1:100) could be observed in cells (presumably neurons) in the dorsal horn. After treatment with PAR4-AP, VSTM2L immunostaining was greatly decreased in the cells in the dorsal horn (Figure 7B), which is in agreement with the proteomics findings. Cell counts in the dorsal horn confirmed that there was a significant difference (*t*-test, *p* < 0.05) between the number of VSTM2L immunopositive neurons in the dorsal horn of scrambled treated mice (control; no pain; 217 ± 14.3 SEM; n = 4) and mice treated with PAR4-AP (persistent bladder pain; 152 ± 8.2; n = 4).

## 3. Discussion

We developed a model of persistent bladder pain in which repeated (3×) activation of intravesical PAR4 receptors resulted in bladder pain (as measured by von Frey threshold sensitivity) lasting to day 9 with little or no evidence of bladder inflammation or injury and no changes in micturition parameters [7]. We consider this to be a useful model to study bladder pain alone while avoiding pain associated with bladder injury and recovery. One limitation of our model is that transurethral catheterization limits studies to females due to the inherent difficulty of male transurethral catheterization without creating urethral and/or bladder damage. Thus, the findings in our model using female mice still have to be validated in a model that includes male mice. However, this model is useful in studying bladder pain, and may mimic the predominant symptom in the majority of IC/PBS patients who do not show bladder pathology [11].

Using this model, we previously reported that intrathecal MIF antagonism reversed established bladder pain [22], strongly suggesting that spinal mechanisms involving MIF are responsible for mediating persistent bladder pain. Our current findings confirm that lumbosacral spinal antagonism of MIF can provide significant relief from persistent bladder pain, with a peak effect at 2 h post-injection (Figure 1B). In agreement with our previous reports [7,22], there were no changes in micturition volume or frequency in this study (Figure 2) and no changes in bladder inflammation/edema (as determined by histology; Figure 3 and Figure 4). Furthermore, we now provide evidence that lumbosacral spinal antagonism of specific MIF receptors (CD74 or CXCR4) can partly reverse persistent bladder pain (Figure 6). Thus, our current findings expand the understanding of MIF-modulated persistent pain by describing the role played by antagonism of individual spinal MIF receptors.

We observed a small yet significant temporary decrease in threshold after intrathecal isotype mAb injection (Figure 5A; see IgG inset) that was not observed after other intrathecal injections of control reagents. It is possible that this was a reaction to the IgG2b; it is also possible that it may represent some minor irritation of the lumbosacral roots due to the intrathecal injection. These mice did not show paraplegia or any other motor deficits after the injections.

We showed that MIF is present in cells (presumably neurons) in the dorsal horn, the intermediolateral area, and the area around the central canal (dorsal grey commissure) [22,25]. These areas are well known to receive bladder afferents that mediate both micturition and bladder nociceptive stimulation [26,27,28,29]. Interestingly, we observed that MIF increased in these areas after persistent bladder pain [22]. MIF, CD74, and CXCR4 have been reported to increase in the dorsal horn in other persistent pain models (neuropathic pain; inflammatory pain) as well [20,30,31].

MIF binds to several receptors, namely, CD74, CXCR2, CXCR4, and CXCR7 [23]. In addition, complexes between these receptors can form and affect signaling [23], indicating that MIF signaling is complex and not explicitly delineated in all systems. The reversal effects observed after individual MIF receptor antagonism in our current study were smaller in magnitude than after blocking MIF (with MIF mAb) intrathecally (Figure 5C). We attribute this to the possibility that, as MIF signaling occurs through multiple receptors, the individual effect of single receptor blockade may be smaller than total MIF antagonism. Future studies could test whether blocking multiple MIF receptors simultaneously results in a larger effect. Alternatively, there may be different doses or different bioavailability parameters with the individual antagonists that could alter effectiveness. Either of these explanations warrants further study to more clearly delineate the mechanisms of MIF-mediated persistent BHA.

Other investigations have implicated MIF in mediating inflammatory and neuropathic pain [19], and a spinal site of action has been identified [20,30]. In addition, a recent study reported that spinal CXCR4 receptors mediate colon inflammation-induced bladder hyperalgesia in rats [32]. Thus, taken together, previous evidence and our current results suggest that spinal MIF and its receptors play a role in modulating spinal mechanisms mediating pain, particularly persistent bladder pain.

In addition, our results show that spinal MIF antagonism resulted in modulation of several spinal proteins (Figure 7). We selected the 2 h post-intrathecal injection time point to examine changes in lumbosacral spinal proteomics, as this was the point of peak analgesic effect from intrathecal MIF antagonism (Figure 1B). We filtered our proteomics findings with a set of criteria (outlined in the Methods section) to allow us to discover MIF-associated protein changes only. We detected decreases in six specific proteins in persistent BHA, and these decreases were reversed by intrathecal MIF antagonism (Figure 6A). MT-1, MT-2, and VSTM2L were similarly identified in our recent study as decreasing during persistent BHA [14]. Metallothioneins are involved in oxidative stress, inflammation [33,34,35,36,37], and possibly pain mediation, although their exact role in pain remains under investigation [38,39]. We recently confirmed using immunohistochemistry that metallothioneins were decreased in the dorsal horn of mice with persistent bladder pain when using a pan-metallothionein antibody [14]. VSTM2L is involved in neuroprotection as well as in neurogenerative and metabolic diseases [40,41,42,43]. Our current findings confirm through immunohistochemistry that VSTM2L decreased in the dorsal horn after persistent BHA (Figure 7), lending further support to our proteomics findings.

We additionally detected MIF-modulated decreases in four spinal proteins. Xaa-pro aminopeptidase 3 (also known as aminopeptidase P3) is part of tumor necrosis factor receptor-2 (TNF-TNFR2) signaling, regulates C-Jun N-terminal kinase (JNK) 1 and 2 phosphorylation, and might have an anti-apoptotic function [44]. Mitochondrial 10-formyltetrahydrofolate dehydrogenase is a substrate of three mitochondrial enzymes that regulate nucleotide synthesis, nicotinamide adenine dinucleotide phosphate (NADPH) formation, and mitochondrial mRNA translation, and may regulate cancer cell metastasis [45]. Methyl-CpG-binding domain protein 3 (MBD3) is part of the methyl-CpG-binding domain family, which interact with methylated CpG dinucleotides and are implicated in neurological disorders and several cancers [46], including liver [47], pancreatic [48], and colon [49]. Interestingly, although we observed a decrease in MBD3 in persistent BHA, a recent study reported no change in spinal mRNA levels of MBD3 in a different model of chronic (neuropathic) pain in rats, although other members of the methy-CpG-binding domain family decreased [50].

Finally, we identified MIF-modulated increases in two specific spinal proteins that were reversed by intrathecal MIF antagonism. Neuronal vesicle trafficking-associated protein 1, also known as NEEP21 (encoded by Nsg1), is an endosomal protein specifically expressed in neurons [51] that plays an important role in the transport of several receptors to regulate synaptic transmission and plasticity [51,52]. Increased Nsg1 expression is found in esophageal squamous cell cancer tissues, where it mediates invasiveness through activation of extracellular signal-regulated kinases (ERK) signaling [53]. This is interesting in light of our current results showing increased NEEP21 levels after persistent BHA and reversal with spinal anti-MIF treatment, as spinal ERK activation mediates bone cancer pain in a rodent model [54]. These findings suggest a possible mechanism to mediate persistent bladder pain. Peptidyl-tRNA hydrolase ICT1 also showed a MIF-modulated increase in persistent BHA. ICT1 is a mitochondrial protein with multiple functions, including protein synthesis and the regulation of cell proliferation and apoptosis [55]. ICT1 is overexpressed in different cancers [55], and while deletion of ICT1 is lethal [56], ICT1 knockdown inhibits proliferation in cancer cells [57].

In conclusion, using a rodent model of persistent bladder pain that focuses on pain as opposed to recovery from bladder injury/pathology, we show that spinal MIF antagonism can reverse persistent bladder pain; spinal MIF receptor antagonism resulted in smaller effects than were seen with MIF antagonism. These findings indicate that spinal changes mediated by MIF and MIF receptors are capable of modulating persistent bladder pain, and increase our understanding of mechanisms of bladder pain in the absence of overt bladder injury. Moreover, our findings using unbiased proteomics uncovered several spinal MIF-modulated proteins that are associated with relief of persistent BHA via spinal MIF antagonism. A direct link from these proteins to MIF and/or pain is absent for many of these proteins; thus, they represent novel spinal MIF-modulated proteins that are involved in persistent bladder pain. The exact mechanism whereby intraspinal MIF antagonism mediates changes in these proteins to result in decreased bladder pain remains to be elucidated, and the roles of spinal MIF and/or MIF receptors in IC/PBS warrant further study to help us understand the mechanisms of persistent bladder pain.

## 4. Materials and Methods

### 4.1. Experiment 1A: Time Course of Analgesic Effect on Persistent BHA after Intrathecal Treatment with Anti-MIF mAb

All animal experiments were approved by the Lexington VA Health Care System Institutional Animal Care and Use Committee (VER-19-005-AF) and performed according to the guidelines of the National Institutes of Health.

We used repeated intravesical instillation of PAR4-activating peptide (PAR4-AP; AYPGKF-NH2; Peptides International, Louisville, KY, USA) to induce persistent BHA as described earlier [7]. Briefly, female mice (C57BL/6; 12–14 weeks of age) were anesthetized with isoflurane (3% induction, 1.5% maintenance) and transurethrally catheterized (PE10, Fisher Scientific, Waltham, MA, USA; 11 mm length). Female mice were used throughout the study because of the relative ease of transurethral catheterization in female mice. Urine was drained by gently applying pressure to the lower abdomen. Bladders were slowly instilled with 100 µL of PAR4-AP (100 µM in sterile PBS; pH 7.4) and remained in the bladder for 1 h. Mice were allowed to recover and returned to their cages. Intravesical treatments were repeated an additional two times at 48 h intervals.

#### 4.1.1. Abdominal Mechanical Sensitivity

Mice were acclimated to the testing conditions as follows.

(Days 1,2) Acclimation to testing room:−Mice placed in testing room and left undisturbed for 3 h(Days 3,4) Acclimation to testing chamber:−Mice placed in testing chamber and left undisturbed for 2 h(Day 5) Acclimation to von Frey monofilaments:−Mice placed in testing chamber and von Frey monofilaments applied to lower abdominal area(Day 7) Baseline von Frey testing - Start of persistent BHA protocol

Testing of lower abdominal mechanical hypersensitivity (interpreted as an index of BHA or bladder pain) was performed as previously described [14,22]. Briefly, 50% mechanical threshold [58] was calculated by measuring the response to calibrated von Frey fibers (0.008, 0.02, 0.07, 0.16, 0.4, 1.0, 2.0 and 6.0 g) applied to the lower abdominal region. A positive response (an indicator of bladder hyperalgesia) was defined as any one of three behaviors: (1) licking the abdomen; (2) flinching/jumping; or (3) abdomen withdrawal. Whenever a positive response to a stimulus occurred, the next smaller von Frey filament was applied. Otherwise, the next higher filament was applied. Thresholds (50%) were measured at baseline (day 0; prior to any treatment) and days 2, 3, 4, 7 and 9.

#### 4.1.2. Optimal Time for Pain Relief after Intrathecal Injection of Anti-MIF mAb

Intrathecal injections were performed as described earlier [22]. Briefly, under isofluorane anesthesia, a 25 µL Hamilton syringe with a 30-gauge needle was inserted between L5 and L6 to reach the intrathecal space as indicated by tail twitch.

On day 7, the optimal time for pain relief after intrathecal injection of either anti-MIF mAb (15 µg; 5 µL; in PBS; n = 6; R. Bucala [59]) or mouse IgG1 (as isotype control; 15 µg; 5 µL; in PBS; n = 6; R. Bucala) was determined by measuring von Frey threshold at 1, 2, 4, and 6 h post-intrathecal injection.

#### 4.1.3. Micturition Parameters in Awake Mice

On day 9, following testing of abdominal mechanical threshold, we measured micturition parameters (micturition volume and frequency) in awake mice using the Voided Stain on Paper (VSOP) method [60] as described earlier [61]. Mice were placed individually in a plastic enclosure that allowed them freedom to move and free access to water. During a 3-h observation period, micturitions were collected on filter paper placed under the enclosure. Using a set of known volumes as a standard, linear regression was used to determine micturition volumes on the filter paper. The micturition frequency indicates the number of micturitions per 3-h observation period.

#### 4.1.4. Histological Measurements

At the end of the study (day 9) mice were anesthetized with 3–4% isofluorane, bladders were rapidly removed, and segments were placed in 4% buffered formaldehyde (Fisher Scientific; #SF100-4) for histology. Bladder paraffin sections (5 µm) were processed for routine hematoxylin and eosin (H&E) staining. The stained sections were evaluated by two independent observers blinded to the assignment of experimental groups and scored separately for edema and inflammation according to the following scale, as described in [7]: 0, no edema/no infiltrating cells; 1, mild submucosal edema/few inflammatory cells; 2, moderate edema/moderate number of inflammatory cells; 3, frank edema, vascular congestion/many inflammatory cells.

### 4.2. Experiment 1B: Effect of Intrathecal Antagonism of Specific MIF Receptors on Persistent BHA

We examined the effect of intrathecal antagonism of CD74, MIF’s cognate receptor [62], as well as the effect of intrathecal antagonism of CXCR4, another functional MIF receptor [63].

Female mice received repeated intravesical instillation of PAR4-AP as described above to set up persistent bladder pain; the protocol was modified to reduce the number of mice used in this study, as was done in an earlier protocol [22], with each group receiving intrathecal antagonist and control treatment as follows:Spinal antagonism of CD74 (n = 6)−On day 7, 15 µg of anti-CD74 (Biolegend; 5 µL) were injected and von Frey measurements were taken at 0, 1, 2, 4, and 6 h post-injection−On day 9, 15 µg of isotype IgG (rat IgG2b; R&D System; MAB0061; 5 µL) were injected and von Frey measurements were taken at 0, 1, 2, 4, and 6 h post-injectionSpinal antagonism of CXCR4 (n = 6)−On day 7, 1 µg of AMD3100 (CXCR4 antagonist; Tocris; #3299; 5 µL) was injected and von Frey measurements were taken at 0, 1, 2, 4, and 6 h post-injection−On day 9, PBS (Vehicle control; 5 µL) was injected and von Frey measurements were taken at 0, 1, 2, 4, and 6 h post-injection

Mice were euthanized at the end of the experiments (day 9).

### 4.3. Experiment 2A: Lumbosacral Spinal Proteomics Changes in Persistent Bladder Pain: Effect of Spinal MIF Antagonism

Female mice received repeated intravesical instillation of PAR4-AP as described above to set up persistent bladder pain. Repeated instillation with a scrambled peptide (YAPGKF-NH2; Peptides International, Louisville, KY, USA) served as a “no pain” control group [7].

On day 7, mice were divided into four groups (n = 9/group):Group 1: Scrambled peptide + sham lumbosacral intrathecal injection (control; no pain group)Group 2: PAR4-AP peptide + sham lumbosacral intrathecal injection (pain group + sham treatment)Group 3: PAR4-AP peptide + lumbosacral intrathecal anti-MIF mAb injection (pain group + treatment)Group 4: PAR4-AP peptide + lumbosacral intrathecal mouse IgG1 injection (pain group + treatment-control)

At the peak of the analgesic effect (2 h, as determined in Experiment 1A), mice were re-anesthetized with isofluorane, the lumbosacral spinal segments were exposed by laminectomy, and the spinal cord was removed, snap-frozen, and stored in liquid nitrogen until analysis.

#### 4.3.1. Proteomics Sample Preparation

Spinal cord tissue was trimmed and submerged in a protein lysis buffer consisting of 1% sodium dodecyl sulfate (SDS) in 1X PBS with protease inhibitor. Repetitive freeze-thaw cycles (three times) were performed using ethanol and dry ice baths within a Styrofoam box (for freezing) and a temperature-controlled thermomixer (for thawing). A tip sonicator was used to homogenize samples on wet ice for 10–15 s (0.3 s ON/0.7 s OFF pulse). Samples were centrifuged at 13,000 rpm for 30 min at 4 °C, and the resulting supernatant containing total proteins was transferred to a newly labeled Eppendorf tube. Total protein concentration was measured using a standard BCA assay kit from Pierce (ThermoFisher, Waltham, MA, USA, 23225). Equal amounts of protein were drawn from each sample, ensuring a similar volume for the analysis. Protein samples were reduced by adding 10 mM dithiothreitol (DTT, Sigma, Livonia, MI, USA, D9779), followed by 30 min incubation at 56 °C in a thermomixer set at 500 rpm. Subsequently, samples were alkylated by 25 mM iodoacetamide (IAA, Sigma, I6125) and incubated for 30 min at 37 °C in the dark. Reduced/alkylated proteins were precipitated by adding pre-chilled acetone and incubating at −20 °C overnight, achieving a protein pellet at the base of the Eppendorf tube after centrifugation at 20,000× *g* for 30 min at 4 °C. The supernatant was removed and the protein pellet was washed with a mixture of one volume of 5:1 acetone to water. The washed pellets were suspended in a digestion buffer (50 mM Triethylammonium bicarbonate, Sigma, T7408, pH 8.0). Enzymatic digestion was carried out by adding Trypsin/Lys-C protease mix (ThermoFisher, A41007) to each sample at a 1:50 ratio and the samples were incubated for 4 h at 37 °C with 500 RPM shaking. This was followed by overnight digestion after the addition of one more part of the enzyme. The digested protein samples were frozen at −80 °C until the tandem mass tag (TMT) labeling step [14].

#### 4.3.2. TMT Labelling and Fractionation

A a 6-plex TMT kit purchased from ThermoFisher (90061) was used. Prior to labelling, TMT labels were hydrated by adding 41 µL of anhydrous acetonitrile and incubated for 5 min at room temperature with intermittent vortexing. An individual sample was labeled with the assigned TMT label (between TMT126-TMT132) and incubated for 2 h at room temperature; 5% Hydroxylamine was added for 15 min to stop the TMT labeling reaction. Quenched individual samples were pooled from respective sets. Pooled sets were further cleaned with Strata-X-C columns (8BS029-TAK-TN) using wash solution (30%MeOH + 0.1% formic acid) and further eluted using elution solution (30% MeOH + 5% ammonium hydroxide). Eluted pools were fractionated using an Ultimate-3000 high-performance liquid chromatography system (Dionex, Sunnyvale, CA, USA). Samples were first trapped in a pre-column (3 µm, C18 particles) and peptides were eluted through a 50 cm EasySpray PepMap RSLC C-18 column (Thermo Scientific, Bremen, Germany). The peptide fractions were acquired using a high-resolution quadrupole orbitrap Q-Exactive mass spectrometer (Thermo Scientific, Bremen, Germany) in a data-dependent mode where the top twelve most intense peptide ions were selected in full MS mode and further fragmented in MS/MS mode. The full MS was acquired with the orbitrap set at 70,000 resolution power (scan range 300–1600 *m/z*); the MS/MS was performed by collision-induced dissociation and acquired at 35,000 resolution (scan range of 100–2000 *m/z*) [14].

#### 4.3.3. Protein Identification and Quantification

Analysis for protein identification and quantification was performed using Proteome Discoverer (Version 2.2.0.388, Thermo Fisher Scientific, Waltham, MA, USA). The spectral data were compared with the Mus musculus protein dataset and with known human contaminants. A multistep analytical workflow consisting of Spectrum Files (data input), Spectrum Selector (spectrum and feature retrieval), Percolator (peptide spectral matching or PSM validation), and Reporter Ions Quantifier (quantification) was created. In all analyses, TMT 6-plex options were selected for all nodes. Only proteins with high or medium confidence (corresponding to a *p*-value ≤ 0.05) were included in subsequent analyses. The results from Proteome Discoverer were exported and unique peptides and all of their quantification values were appended to the corresponding protein accession numbers [14].

### 4.4. Experiment 2B: Preliminary Validation of Targets Using Immunohistochemistry

We replicated part of the experimental paradigm as described, including two only groups (n = 4/group): Group 1: Intravesical scrambled peptide + sham i.t. (no pain); Group 2: Intravesical PAR4-AP + sham i.t. (persistent BHA; unalleviated pain group). Two hours after intrathecal injections, the mice were transcardially perfused with saline followed by 4% paraformaldehyde. L6/S1 spinal cord segments were excised, placed in 4% paraformaldehyde overnight, and then processed for paraffin embedding.

Paraffin sections (5 µm) of paraformaldehyde-fixed L6/S1 spinal cord were dewaxed and processed for antigen retrieval using citrate buffer (pH 6.0) at 94 °C for 20 min. Sections were treated with 3% H_2_O_2_ for 10 min to quench endogenous peroxidase and then with normal donkey serum for 1 h followed by incubation with primary antibody (mouse anti-VSTM2L; Santa Cruz, Santa Cruz, CA, USA; LO55, at 1:100 overnight at 4 °C). Bound primary antibody was detected using a VECTASTAIN^®^ ABC Kits (PK-4001, Vector Labs, Newark, CA, USA). Digital images were captured with a Leica Microscope (DMI4000B, Wetzlar, Germany). One spinal cord section from each mouse was selected at random and VSTM2L-positive cells were counted in each dorsal horn.

### 4.5. Statistical Analysis

Statistical analysis was performed using R [64]. For experiments 1A and 1B, changes in the von Frey threshold after intrathecal injection were analyzed using repeated measures analysis of variance (ANOVA) and, if significant, Dunnett’s tests comparing the von Frey threshold at different time points to 0 (baseline). In experiment 1A, changes in the micturition parameters and histological scores were analyzed using Student’s *t*-test, while changes in inflammation and edema score were assessed using Fisher’s exact test. In experiment 1B, peak effect comparisons between all three intrathecal treatments were analyzed using ANOVA and, if significant, Dunnett’s test, using anti-MIF as the comparison group. For experiment 2B, differences in the number of immunopositive neurons were assessed using a *t*-test. Data are reported as mean ± standard error of the mean (SEM) or median and interquartile range, as appropriate.

In experiment 2A, TMT sets were compared using ratios between samples and pools and the data were transformed to log2 scale. A minimum of 50% of missing values was considered when analyzing proteins. In order to determine whether any significant differences existed between groups of interest, ANOVA was performed for each protein. The False Discovery Rate (FDR) was used to correct *p*-values based on differences between TMT sets. A post hoc test (*t*-test) was conducted by comparing estimated marginal means from all possible contrasts (pairs of groups). To determine MIF-modulated proteins, we selected proteins that showed significant changes compared to control (no pain group) after intravesical PAR4-AP (sham or IgG1 i.t. treatment) and that showed no significant change after i.t. MIF antagonism. Protein levels are displayed as the ratio score normalized to control and converted to log2 scale.

## 5. Conclusions

Spinal MIF and the MIF receptors CD74 and CXCR4 mediate persistent bladder pain and spinal proteomics changes. The exact mechanisms and the clinical relevance to persistent pain conditions (including IC/PBS) remain to be determined.

## Figures and Tables

**Figure 1 ijms-25-04484-f001:**
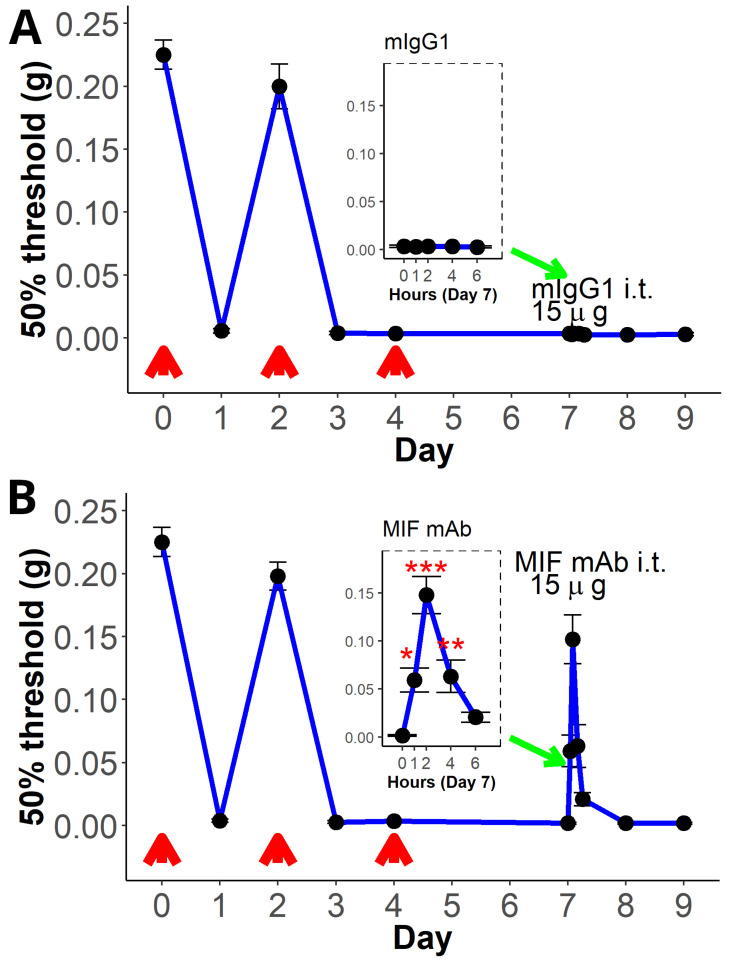
Effect of intrathecal MIF antagonism on persistent bladder pain. The 50% threshold to von Frey stimulation is depicted on the y-axis, while the time in days is depicted on the x-axis. Insets (green arrows) show the time in hours post-intrathecal injection. Differences in threshold after treatment were determined using repeated measures ANOVA followed by Dunnett’s post hoc tests (compared to time 0). Repeated (3×; red arrows) intravesical instillation of PAR4-AP resulted in marked decrease in threshold (an index of bladder pain) that recovered on day 2 but remained decreased after subsequent instillations and persisted until day 9. This is our index for persistent bladder pain. The effects of intrathecal (L6/S1) injections on day 7 were assessed using repeated-measures ANOVA followed by post hoc Dunnett’s tests comparing all the time periods to 0. (**A**) Intrathecal injection of isotype control had no effect on decreased von Frey threshold, and consequently no effect on established persistent BHA. (**B**) Intrathecal injection of a neutralizing MIF mAb quickly reversed established persistent BHA (as shown by the increased threshold), with a maximal effect at 2 h post-injection. * = 0.05, ** = 0.01, *** = 0.001. The insets show the von Frey threshold for up to 6 h after i.t. injection.

**Figure 2 ijms-25-04484-f002:**
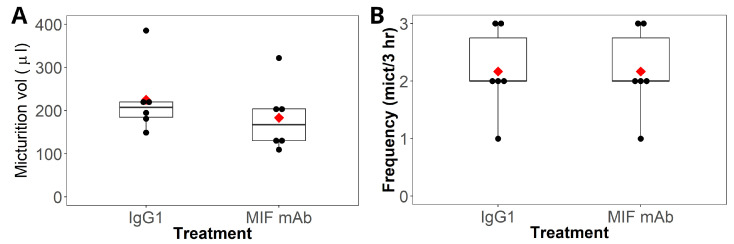
Boxplot showing the median and interquartile range (the mean is displayed as the red diamond) and dotplot of the micturition volume and frequency for each group as evaluated on day 9 of the protocol. Differences between the two groups were evaluated using *t*-tests. (**A**) The micturition volume (in µL) shows no significant difference between the two groups (*p* = 0.40). (**B**) The micturition frequency (number of micturitions/3 h) is identical between the two groups; therefore, no significant difference was observed (*p* = 1).

**Figure 3 ijms-25-04484-f003:**
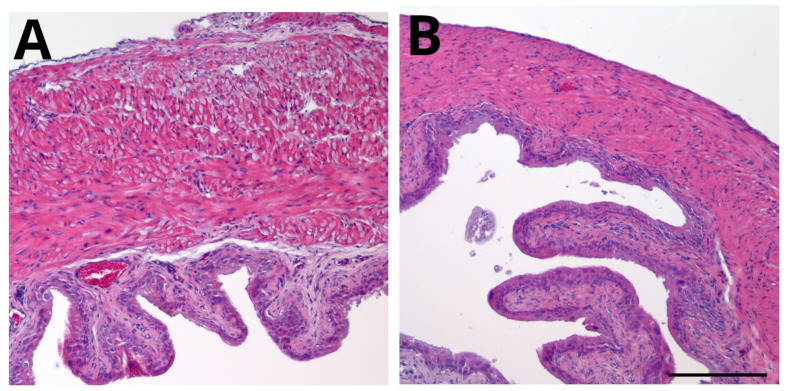
Histological examination of bladder inflammation and edema. Bladders were collected on day 9 (end of experiment), processed for H&E staining, and scored for inflammation and edema as listed above. Representative sections are presented for each group. There is little or no evidence of urothelial damage and/or suburothelial inflammation and/or edema in mice from either group. (**A**) Mice treated with 15 µg IgG1 (isotype control) intrathecally. (**B**) Mice treated with 15 µg anti-MIF monoclonal antibody intrathecally. Calibration bar = 200 µm.

**Figure 4 ijms-25-04484-f004:**
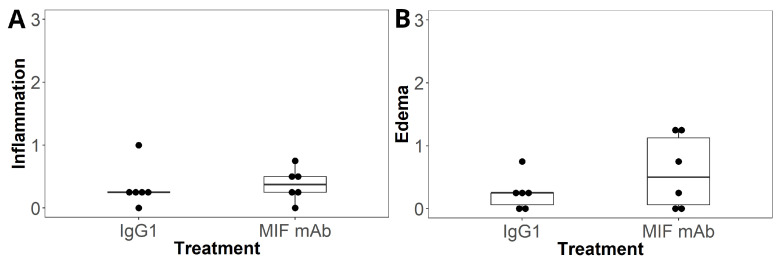
Boxplots showing median and interquartile range along with dotplots for inflammation and edema scores. Differences between the two groups were assessed using Fisher’s exact test. (**A**) There is no statistical difference (*p* = 0.52) in inflammation scores between mice treated with intrathecal IgG1 vs intrathecal MIF neutralizing antibody (MIF mAb). (**B**) Edema scores between the two groups do not show a statistically significant difference (*p* = 0.71).

**Figure 5 ijms-25-04484-f005:**
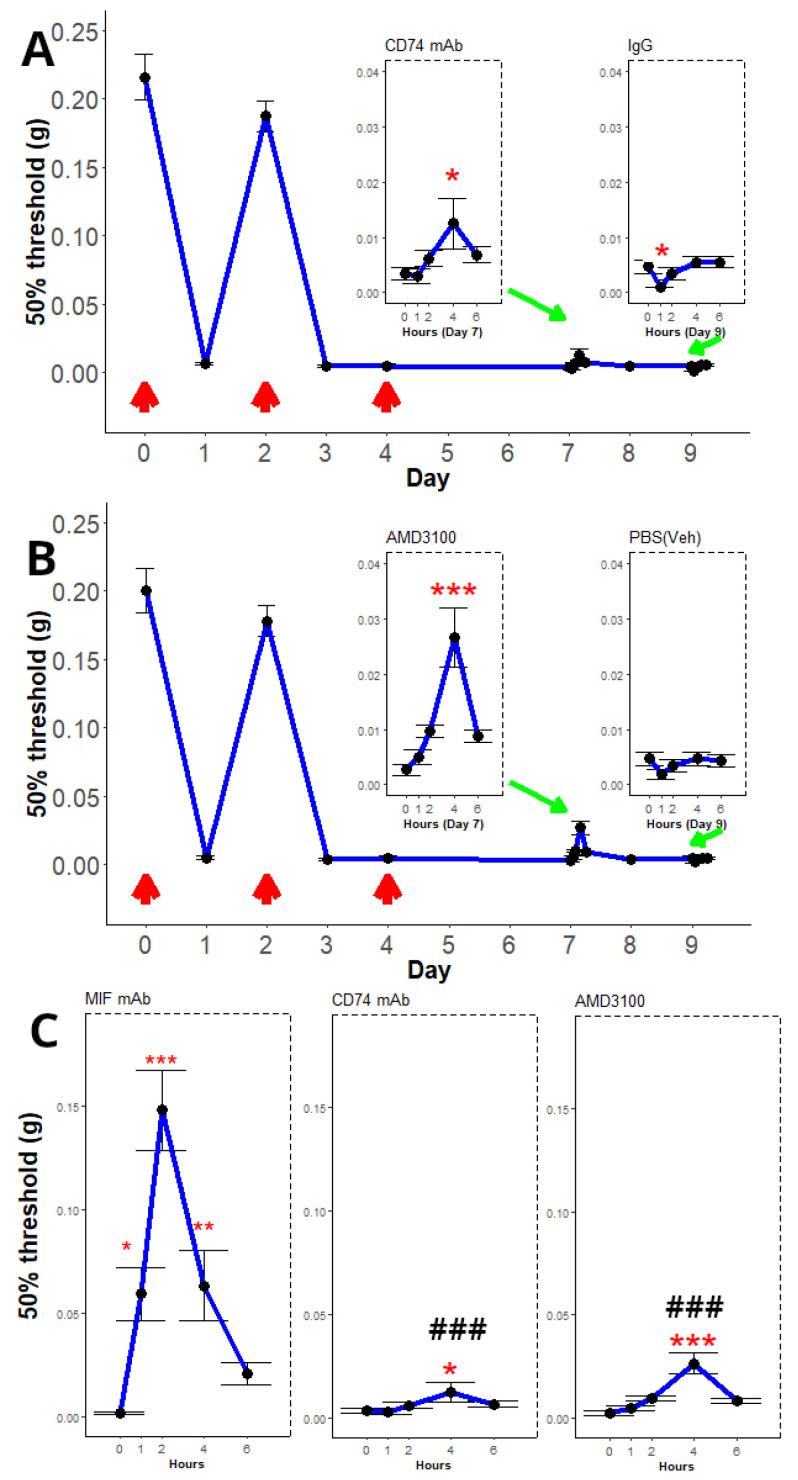
Effect of intrathecal antagonism of MIF receptors on persistent bladder pain. The 50% threshold to von Frey stimulation is depicted on the y-axis, while the time in days is depicted on the x-axis. Red arrows indicate days of intravesical PAR4 installation. Insets (green arrows) show the time in hours post-intrathecal injection. Differences in threshold after treatment were determined using repeated measures ANOVA followed by Dunnett’s post hoc tests (compared to time 0). (**A**) Intrathecal treatment with anti-CD74 mAb had a small yet significant (* = *p* < 0.05) reversal of persistent BHA at 4 h post-injection. Intrathecal treatment with rat IgG2b (isotype control) showed a small significant decrease in threshold at 1 h (* = *p* < 0.05). (**B**) Intrathecal administration of AMD3100 (CXCR4 antagonist) on day 7 partially reversed persistent BHA, with the peak effect occurring 4 h after treatment (*** = *p* < 0.001). Intrathecal treatment with PBS (vehicle control) on day 9 had no effect on persistent BHA (*p* = 0.29). (**C**) The effects of the three spinal antagonists (MIF mAb, CD74 mAb, and AMD3100) were evaluated using ANOVA followed by Dunnett’s post hoc tests. Both CD74 mAb and AMD3100 showed significant differences (### = *p* < 0.001) when compared to the response to MIF mAb. * = *p* < 0.05, ** = *p* < 0.01, *** = *p* < 0.001 when compared to 0 hours.

**Figure 6 ijms-25-04484-f006:**
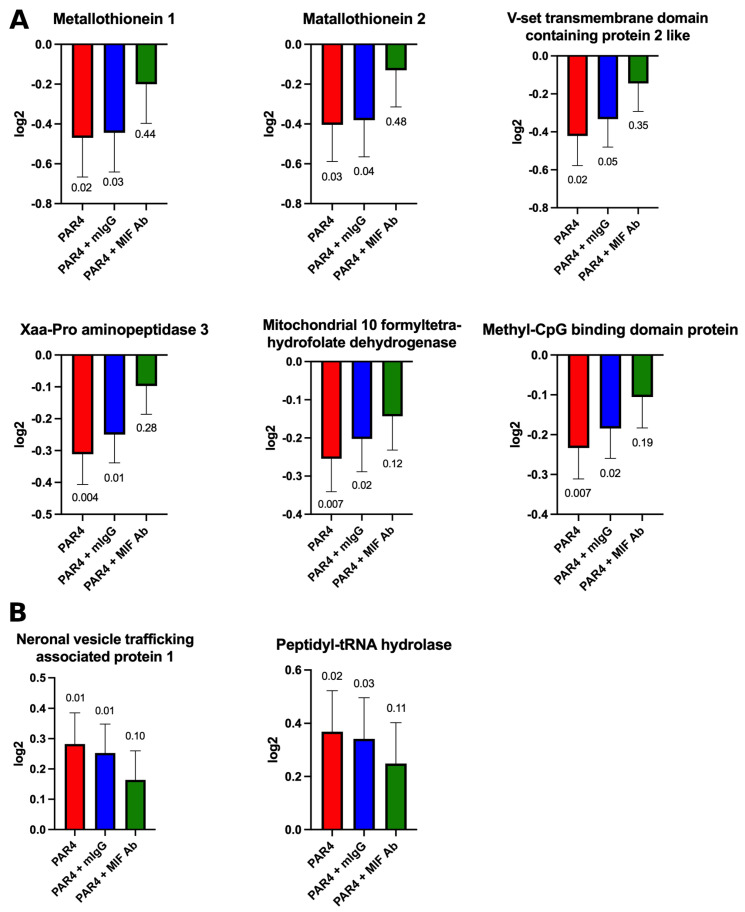
Protein levels were compared using ratios between samples, then pools and data were transformed to log2 scale (y-axis). ANOVA was used to determine whether any significant differences existed between the groups. Post hoc tests (*t*-test) were used to compare the estimated marginal means between all possible contrasts (pairs of groups). Because all ratios are normalized to control, the protein ratio for the control group is not shown. Each panel shows the estimated marginal means and standard error of the mean for each group along with the *t*-test significance compared to control. Novel MIF-modulated proteins were determined by selecting proteins in the persistent (sham i.t.; Group 2; red) and unalleviated BHA (isotype IgG1 i.t; blue, Group 3) groups that showed a significant difference when compared to control group (no pain; Group 1) and no significant difference when MIF mAb i.t. treatment was compared to control. (**A**) Persistent and unalleviated BHA (red; blue) showed decreased levels in six proteins depicted and increased levels in two proteins (**B**). These changes were reversed by intrathecal treatment with a MIF mAb (green; Group 4). All ratios are normalized to the control group.

**Figure 7 ijms-25-04484-f007:**
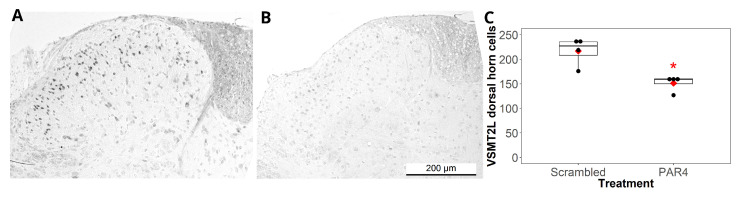
Decreased V-set transmembrane domain-containing protein 2-like (VSTM2L) immunostaining in L6/S1 spinal cord in the persistent BHA group. (**A**) Representative image of VSTM2L immunostaining image from a dorsal horn of a mouse treated with scrambled peptide (no pain). Positive cells (presumably neurons) can be seen in the dorsal horn. (**B**) Representative image of VSTM2L immunostaining in the dorsal horn area from a mouse treated with repeated instillations of PAR4-AP to induce persistent BHA. A significant decrease in staining of the cells in the dorsal horn support the findings from the proteomics analysis. (**C**) For each spinal cord section, immunopositive neurons in both dorsal horns were counted and summed; a boxplot showing the median and interquartile range (with the mean displayed by the red diamond) along with a dotplot is presented for both treatment groups. There was a significant decrease (* *p* < 0.05) in the number of VSTM2L positive cells in the PAR4-AP (PAR4) group.

## Data Availability

The raw data supporting the conclusions of this article will be made available by the authors on request.

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
