# Peer review of "MIF-Modulated Spinal Proteins Associated with Persistent Bladder Pain: A Proteomics Study"

_ijms, 2024, doi:10.3390/ijms25084484_

Round 1

Reviewer 1 Report

Comments and Suggestions for Authors

I want to congratulate the authors for a very well written and interesting manuscript. The way the model for bladder pain was constructed was very interesting. There are some issues in the article that must be addressed.

Line 102: what is the reason the reference 26 was inserted? Please review and explain.

Line 258: the text reads "The 50% threshold recovers after the first instillation and then remains decreased...". This is a somehow misleading. If it remains means it stays at the previous level which was increased. In reality the threshold decreases again after temporarily recovering. Please review and if necessary correct.

Figure 3: The magnifications on image A and image B are the same? Please comment.

The conclusions section must describe more faithfully the data described in the results section (e.g. The analysis demonstrated smaller effects for CD74 and CXCR4 antagonists than anti-MIF CD74 mAb)

Author Response

Dear Dr Tancharoen,

Thank you for the review of our manuscript titled “MIF-modulated spinal proteins associated with persistent bladder pain: a proteomics study”  for consideration for publication in the International Journal of Molecular Sciences as part of a special edition on “New Insights into the Molecular Mechanisms of Chronic Pain”.

We thank the reviewers for the thoughtful and thorough review. We addressed the comments and outlined below and we believe the manuscript has improved as a result of the reviewers’ suggestions. We are submitted a marked-up pdf with our changes, a revised Figure 7 and a clean latex version of the manuscript. Please note: Text deletions are marked as strike-out red text. Text additions are underlined and in blue color. All page and line references below correspond to the marked up copy of the manuscript.

We made the following changes:

“Line 102: what is the reason the reference 26 was inserted? Please review and explain.”

Reference was inserted to show that the antibody from Dr. Bucala has documented anti-MIF effects in vivo.

“Line 258: the text reads "The 50% threshold recovers after the first instillation and then remains decreased...". This is a somehow misleading. If it remains means it stays at the previous level which was increased. In reality the threshold decreases again after temporarily recovering. Please review and if necessary correct.”

We agree the wording was confusing. We reworded the sentence now found on p. 6, l. 261-263 of marked up copy provided.

“Figure 3: The magnifications on image A and image B are the same? Please comment.”

The magnifications are the same (100x). We selected sections that were representative of the reported median score. The difference is size is likely the result of a slight misalignment of the bladder specimen during paraffin block formation.

“The conclusions section must describe more faithfully the data described in the results section (e.g. The analysis demonstrated smaller effects for CD74 and CXCR4 antagonists than anti-MIF CD74 mAb)”

This is an excellent point, and we revised our wording to reflect this point as shown in p. 18, l. 451-453.

Reviewer 2 Report

Comments and Suggestions for Authors

The study by Ye and coworkers focused on bladder pain, a traditionally difficulty area of research. They developed a model of bladder pain without the presence of signs of bladder inflammtiona, which the authors propose is likely to replicate the syndrome of IC/BPS without Hunner's lesions. The authrs found that intravesical activation of PAR-4 leads to heightened abdominal cutaneous sensitivity to mechanical stimulation, which was interpreted as bladder pain. Animals did not present any signs of bladder overactivity or inflammatory signs. Intrathecal administration of an anti-MIF antibody improved pain. This was also observed after adminstration of antagonists of MIF receptors but was less evident. Proteomic analysis showed that MIF-related bladder pain was accompanied by differential expression of specific proteins, likely involved in the establishment of chronic bladder pain.

Overall, this is an interesting study that expands on previous studies from the same research team. There are a few points that require the attention of the authors:

- on page 2: I assume that the habituation to sensory testing was performed before intravesical treatment. Ho many days before?

- on page 3: how were the intrathecal injections performed? With catheters? by direct puncture? At which level?

- on page 3, line 131; on page 4, line 130: why not separate grousp and rather treat the same animals with the drug followed by the vehicle?

- on page 10: why better results with the anti-MIF antibody than with the antagonists? Which cells are being targeted by the anti-MIF antibody or the receptor antagonists? Why did the authors not test a combination of the antagonists? Are these the only MIF receptors?

- Figure 7: why did the authors prefer light microscopy? Immunofluorescence would have allowed double staining, for example, with NeuN to confirm the expression of VSTM2L in neurons. Can the image be improved in terms of colour/contrast?

General comments:

While I agree that there is evidence that MIF is involved in the central neuronal mechanisms underlying bladder pain, it is not clear from this study (nor included in the Discussion) which cells express MIF, which cells MIF receptors, if MIF colocalizes with its own receptors and if MIF or its receptors colocalizes with the proteins identified in the proteomic assays.

Comments on the Quality of English Language

Good use of English.

Author Response

Dear Dr Tancharoen,

Thank you for the review of our manuscript titled “MIF-modulated spinal proteins associated with persistent bladder pain: a proteomics study”  for consideration for publication in the International Journal of Molecular Sciences as part of a special edition on “New Insights into the Molecular Mechanisms of Chronic Pain”.

We thank the reviewers for the thoughtful and thorough review. We addressed the comments and outlined below and we believe the manuscript has improved as a result of the reviewers’ suggestions. We are submitted a marked-up pdf with our changes, a revised Figure 7 and a clean latex version of the manuscript. Please note: Text deletions are marked as strike-out red text. Text additions are underlined and in blue color. All page and line references below correspond to the marked up copy of the manuscript.

We made the following changes:

Reviewer 2

“- on page 2: I assume that the habituation to sensory testing was performed before intravesical treatment. Ho many days before?”

Yes. Habituation was indeed performed before started the intravesical treatments. The Days listed in the habituation description present a timeline and we have added that on day 7, after the baseline VF testing, we start intravesical treatment to induce persistent BHA (p. 3, l. 90).

“- on page 3: how were the intrathecal injections performed? With catheters? by direct puncture? At which level?”

We added more details to our intrathecal injection procedures, which was done by a lumbar puncture between L5-L6 (p. 3, l. 101-103).

“- on page 3, line 131; on page 4, line 130: why not separate groups and rather treat the same animals with the drug followed by the vehicle?”

For these experiments, each group received i.t. treatment (day 7) and control i.t. injection on (day 9). This was done to reduce the number of animals used and it’s a protocol that we have used before. This is now stated clearly in on p. 3; l. 132-133.

“- on page 10: why better results with the anti-MIF antibody than with the antagonists? Which cells are being targeted by the anti-MIF antibody or the receptor antagonists? Why did the authors not test a combination of the antagonists? Are these the only MIF receptors?”

We hypothesized in the discussion that since MIF binds to multiple receptors (including CD74 and CXCR4 tested in this study) it is possible that antagonism of a single receptor may result in a smaller effect than antagonism of the common ligand (e.g., MIF) for each receptor. We have expanded that section of the discussion to list all (so far) known MIF receptors (p. 13, l 389-391). We did not test a combination of single receptor blockade to avoid combining Ab and chemical injections. We agree that future studies should examine the effect of single or combined receptor antagonism to better define receptor usage (p. 13, l. 395-396).

“- Figure 7: why did the authors prefer light microscopy? Immunofluorescence would have allowed double staining, for example, with NeuN to confirm the expression of VSTM2L in neurons. Can the image be improved in terms of colour/contrast?”

The aim of the experiment was to verify proteomics findings. As such, immunoperoxidase IHC was used. We agree that future studies should investigate localization. Both panels of Figure 7 were simultaneously adjusted for contrast/light levels. We hope this figure is improved.

“General comments:

While I agree that there is evidence that MIF is involved in the central neuronal mechanisms underlying bladder pain, it is not clear from this study (nor included in the Discussion) which cells express MIF, which cells MIF receptors, if MIF colocalizes with its own receptors and if MIF or its receptors colocalizes with the proteins identified in the proteomic assays.”

We agree that future studies should address and determine the localization of spinal MIF and MIF receptors in bladder pain pathways. The object of the current study was to provide evidence of spinal changes and verification of some of these changes. We provided a brief overview of the location of spinal MIF, CD74 and CXCR4 in the discussion (p. 12, l. 384-385, p13, l. 386-388). Future work should also address a mechanism of how MIF (or MIF receptors) regulate the levels of the proteins listed in this study. There is still work to be done!

Round 2

Reviewer 1 Report

Comments and Suggestions for Authors

I would like to thank the authors for answering to all the queries in a satisfactory manner. The article is worth to be published in the current form.

Author Response

Thank you

Reviewer 2 Report

Comments and Suggestions for Authors

Overall, the authors have improved the manuscript but it is still not clear which cells express the MIF receptors and likely targeted by pharmacological intervention.

Comments on the Quality of English Language

Nothing to point out.

Author Response

I understand and agree with this assessment. It's a worthwhile future study. It was not the intent of the present study.

Thanks.

Round 3

Reviewer 2 Report

Comments and Suggestions for Authors

Nothing else to point.

Comments on the Quality of English Language

Nothing else to point.